# Epidemiology and treatment outcomes of recurrent tuberculosis in Tanzania from 2018 to 2021 using the National TB dataset

**Belinda J. Njiro**[1]*, **Riziki Kisonga**[2], **Catherine Joachim**[3], **Galus Alfredy Sililo**[2], **Emmanuel Nkiligi**[2], **Latifat Ibisomi**[1,4], **Tobias Chirwa**[1], **Joel Msafiri Francis**[5]

**1** Division of Epidemiology and Biostatistics, School of Public Health, Faculty of Health Sciences, University of the Witwatersrand, Johannesburg, South Africa, **2** National Tuberculosis and Leprosy Program, Ministry of Health, Dodoma, Tanzania, **3** Programs and Health Systems Strengthening, Ministry of Health, Dodoma, Tanzania, **4** Nigerian Institute of Medical Research, Lagos, Nigeria, **5** Department of Family Medicine and Primary Care, School of Clinical Medicine, Faculty of Health Sciences, University of the Witwatersrand, Johannesburg, South Africa

* belindaj.njiro@gmail.com

**Data Availability Statement:** There are legal restrictions to sharing the datasets as the datasets are owned by a third party, which is the Ministry of Health, Tanzania. The datasets used and/or

## Abstract

### Background

Patients with recurrent TB have an increased risk of higher mortality, lower success rate, and a relatively feeble likelihood of treatment completion than those with new-onset TB. This study aimed to assess the epidemiology of recurrent TB in Tanzania; specifically, we aim to determine the prevalence of TB recurrence and factors associated with unfavourable treatment outcomes among patients with recurrent TB in Tanzania from 2018 to 2021.

### Methods

In this cross-sectional study, we utilized Tanzania's routinely collected national TB program data. The study involved a cohort of TB patients over a fixed treatment period registered in the TB and Leprosy case-based District Health Information System (DHIS2-ETL) database from 2018 to 2021 in Tanzania. We included patients' sociodemographic and clinical factors, facility characteristics, and TB treatment outcomes. We conducted bivariate analysis and multivariable multi-level mixed effects logistic regression of factors associated with TB recurrence and TB treatment outcomes to account for the correlations at the facility level. A purposeful selection method was used; the multivariable model included apriori selected variables (Age, Sex, and HIV status) and variables with a p-value <0.2 on bivariate analysis. The adjusted odds ratio and 95% confidence interval were recorded, and a p-value of less than 0.05 was considered statistically significant.

### Findings

A total of 319,717 participants were included in the study; the majority were adults aged 25–49 (44.2%, n = 141,193) and above 50 years (31.6%, n = 101,039). About two-thirds were male (60.4%, n = 192,986), and more than one-fifth of participants (22.8%, n = 72,396) were HIV positive. Nearly two in every hundred TB patients had a recurrent TB episode (2.0%, n

analyzed during the current study can be requested by contacting the Permanent Secretary's office at the Ministry of Health Tanzania. The Permanent Secretary's general email address is ps@afya.go.tz.

**Funding:** This research project was supported by a postgraduate training scholarship from the TDR, the UNICEF/UNDP/World Bank/WHO Special Programme for Research and Training in Tropical Diseases - Hosted at the World Health Organization in Geneva, Switzerland, Grant Number B40299 to BJN. The funders had no role in study design, data collection and analysis, decision to publish, or preparation of the manuscript.

**Competing interests:** The authors declare that they have no competing interests

= 6,723). About 10% of patients with recurrent TB had unfavourable treatment outcomes (9.6%, n = 519). The odds of poor treatment outcomes were two-fold higher for participants receiving treatment at the central (aOR = 2.24; 95% CI 1.33–3.78) and coastal zones (aOR = 2.20; 95% CI 1.40–3.47) than the northern zone. HIV-positive participants had 62% extra odds of unfavourable treatment outcomes compared to their HIV-negative counterparts (aOR = 1.62; 95% CI 1.25–2.11). Bacteriological TB diagnosis (aOR = 1.39; 95% CI 1.02–1.90) was associated with a 39% additional risk of unfavourable treatment outcomes as compared to clinical TB diagnosis. Compared to community-based DOT, patients who received DOT at the facility had 1.39 times the odds of poor treatment outcomes (aOR = 1.39; 95%CI 1.04–1.85).

## Conclusion

TB recurrence in Tanzania accounts for 2% of all TB cases, and it is associated with poor treatment outcomes. Unfavourable treatment outcomes were recorded in 10% of patients with recurrent TB. Poor TB treatment outcome was associated with HIV-positive status, facility-based DOT, bacteriologically confirmed TB and receiving treatment at the hospital level, differing among regions. We recommend post-treatment follow-up for patients with recurrent TB, especially those coinfected with HIV. We also propose close follow-up for patients treated at the hospital facility level and strengthening primary health facilities in TB detection and management to facilitate early treatment initiation.

## Author summary

### Why was this study done?

- TB recurrence contributes to TB burden and incidence globally, especially among high TB burden countries. Patients previously treated for TB have a high likelihood of acquiring a recurrent TB episode.

- Recurrent TB is associated with lower cure rates and a high risk of TB drug resistance. A recent systematic review reported successful treatment outcomes in only 68.4% of patients previously treated for TB.

- In Tanzania, it was reported in 2018 that patients who had previously undergone TB treatment had an approximate 89% success rate, with 6.6% of those who had recurrent TB dying during treatment. The necessity for more studies in this specific re-treatment group is driven by the absence of evidence regarding the treatment outcomes for patients with recurrent TB.

### What did the researchers do and find?

- We analysed a national dataset of all patients with TB diagnosis from 2018 to 2021 recorded in the DHIS2-ETL database countrywide. We determined TB treatment outcome as either favourable if the patients were considered cured or completed treatment; or unfavourable if they were lost to follow-up (default), with treatment failure, or died.

We established possible determinants for poor treatment outcomes and considered both individual and facility-level effects in the analysis through a multilevel regression model.

- About 10% of patients with recurrent TB had unfavourable treatment outcomes; death was the most reported poor outcome affecting 6% of recurrent TB patients. Patients coinfected with HIV, those treated under facility-based DOT, and patients who received treatment in Zanzibar, Coastal, and Central geographical zones had higher rates of poor outcomes. Patients with bacteriologically confirmed TB and who were treated at the hospital were more likely to have unfavourable treatment outcomes.

### What do these findings mean?

- There is a need to design and implement interventions that are specifically targeted for managing patients with TB recurrence, especially for HIV coinfected patients.

- Drug susceptibility testing and close monitoring after treatment completion are crucial to prevent recurrence. Also, ensuring early detection and treatment and promoting short- and long-term improvements in treatment outcomes for these patients.

- Capacitating and strengthening Primary health care facilities for TB diagnosis and treatment may be a promising approach to promote early TB detection and treatment initiation and subsequently maintain better outcomes for patients with recurrent TB. This should be coupled with close monitoring of patients treated at the hospital level through an appropriate DOT strategy.

### Introduction

Despite improvements in global control tactics, tuberculosis still poses a threat to global health, particularly in lower- and middle-income nations. It is the leading cause of infectious disease-related death worldwide [1,2]. The World Health Organization (WHO) reported a total of 10.6 million cases of TB in 2021 globally [1]. The burden of TB is highest in Southeast Asia and the WHO African region, with these regions harbouring about 70% of all TB cases. Additionally, the African region accounts for over 50% of the HIV/TB burden. Among others, the high burden of TB is attributed to the high HIV burden in this region [3].

TB recurrence comprises a subsequent TB diagnosis in patients previously treated for TB and declared cured or completed treatment [4]. This can be attributed to a reactivation of the previously treated TB disease (relapse) or reinfection with a new MTB strain [5]. The risk of developing a recurrent episode of TB among patients with previous confirmed TB diagnosis ranges between 8% and 10% in 1–2 years after treatment; this risk is higher among HIV-infected persons [5,6]. A systematic review reported a pooled recurrent TB incidence of 2.26 per 100 person-years, with a rate of 4.1 per person-years reported in high TB incidence settings. A study conducted in India reported a 47% proportion of TB recurrence among patients with drug-resistant TB [7].

Tanzania is included in the WHO global list of countries reported to have high TB burden and high TB/HIV coinfection worldwide [8]. TB notification in Tanzania has been shown to

increase between 2019 to 2021. Among these, the proportion of TB relapse in 2020 was reported at two percent, and the overall success rate for both new and relapse TB patients is 94% [9].

Studies report the role of high HIV burden on TB recurrence, with HIV patients having a higher risk of a recurrent TB episode [10]. Previous investigations have identified a wide range of other risk factors for recurrence, including demographic, socioeconomic, clinical, and bacteriologic variables. [6]. Increased risk of TB recurrence has also been reported among patients with a diagnosis of drug-resistant TB, smear-positive TB disease [11], chronic lung disease, and history of smoking and substance use [12]. Understanding the determinants of recurrent TB may enable health professionals and control efforts to identify vulnerable individuals who are more susceptible to TB recurrence [6].

TB recurrence poses a threat to the TB control program, especially in high TB and HIV burden countries. Lower cure rates and poor treatment outcomes have been reported among TB retreatment cases in diverse settings. In lower- and middle-income countries (LMICs), patients with TB relapses and treatment after failure were less likely to have successful outcomes, with success rates ranging from 54% to 77% [13,14]. Further, TB recurrence poses a risk of anti-TB drug resistance; TB relapse patients with poor outcomes are more likely to harbour and transmit drug-resistant TB from previous treatment [15]. Recurrent TB was also linked with a high mortality rate, with over a quarter of participants dying during treatment [16].

TB recurrence imposes long-term health complications for patients and additional costs in the treatment [17]. It also poses a threat to the development of drug-resistant TB [15]. In addition to the contribution of TB recurrence in incident TB cases in Tanzania, studies reporting treatment outcomes for patients with recurrent TB are scarce and not representative of the general Tanzanian population. Analysing and reporting treatment outcomes for patients with TB recurrence is crucial in monitoring the progress of the TB control program in Tanzania. Further, identifying patients' and facility characteristics associated with a higher risk of poor treatment outcomes will inform context-specific control strategies to ensure efficacious treatment and reduce TB transmission, morbidity, and mortality. We, therefore, analysed national four-year TB data to assess the epidemiology of recurrent TB in Tanzania; specifically, we aim to determine the prevalence of TB recurrence and factors associated with unfavourable treatment outcomes among patients with recurrent TB in Tanzania from January 2018 to December 2021.

## Methods

### Ethical considerations

Permission to use the DHIS2-ETL TB data was sought from the National TB and Leprosy program, Ministry of Health, Tanzania. We obtained ethical clearance from the University of the Witwatersrand Research Ethics Committee (Medical)–ethics clearance certificate number M221181. An ethical clearance waiver was granted from the Tanzania National Research Ethics Committee (NatREC) at the National Institute of Medical Research (NIMR); reference number NIMR/HQ/R8a VOL VII/30. Privacy and confidentiality of the data obtained were closely observed by encrypting the dataset and ensuring access of data only to authorized investigators. To ensure the anonymity of the study participants, deidentification was done by the TB program personnel before the data was shared. Additionally, the data was used in codes, and no identifying information was available in the database. The data will be stored in Stata format files for five years after dissemination.

## Study design

This is a cross-sectional study of a cohort of TB patients treated over a fixed length period from the Tanzania National TB dataset. We conducted secondary data analysis of the individual case data in the district health information software version 2 (DHIS2-ETL) database using routinely obtained data from the Tanzania National TB Program dataset. The DHIS2-ETL was first rolled out in January 2018; it is a web-based system that captures patients' socio-demographic and relevant clinical characteristics as well as diagnostic and laboratory test results of patients with TB in Tanzania. Relevant clinical details of patients diagnosed and initiated TB treatment are recorded in a paper-based TB register at the respective TB facilities. The data is then subsequently entered into the DHIS2-ETL database.

## Study setting

TB individual patient data from all 31 regions of Tanzania mainland and Zanzibar were utilized. In 2021, the estimated population of Tanzania was projected to be 57,724,380, according to the 2012 National Census Data [18]. There were a total of 8,458 health facilities providing services in the country; these include 7,200 dispensaries, 926 health centres, and 369 hospitals [19]. In 2018, a total of 1,613 facilities were designated TB diagnostic centres providing at least smear microscopy laboratory services [20]. The annual TB notification rate in Tanzania increased from 145 to 148 per 100,000 general population between 2019 and 2020 [21].

## Study population

The national TB dataset from 2018 to 2021 contained information for both adults and children with confirmed TB diagnoses. For this study, all patients in the dataset were included in the analysis.

## Sampling and sample size

We calculated the minimum sample size and statistical power to estimate the prevalence of unfavourable outcomes for patients with recurrent TB using Open Epi Source [22]. The National TB program reports a treatment success rate of 94% among TB cases in Tanzania (unexposed group), with a 6% unsuccessful treatment rate [23]. We hypothesized a higher proportion of unfavourable or unsuccessful treatment outcomes among patients with recurrent TB. For the exposed group (TB recurrent patients), we used the 54% treatment success rate among recurrent TB patients reported in the Uganda [13]. Using HIV status as the key exposure indicator, the prevalence of successful outcomes for HIV-positive and HIV-negative recurrent TB patients was 52% and 38%, respectively, with the effect size (Odds ratio) of 1.77. For investigating factors associated with unfavourable outcomes, we obtained a minimum sample size of 434 to attain 80% statistical power at a 5% significance level.

## Study variables

**Outcome variables.** In this study, the outcomes of interest were TB recurrence among TB patients attended from 2018 to 2021 and TB treatment outcomes of patients with recurrent TB. TB recurrence was the primary outcome variable, defined by WHO as patients recorded with new TB episodes after previously undergoing treatment for TB and declared cured or had finished their therapy at the end of the most recent cycle. The secondary outcome variable was unfavourable treatment outcomes of patients with recurrent TB. According to WHO, TB treatment outcomes are categorized into 5 groups: cured, completed treatment, treatment failure, loss to follow-up, and death. In this study, TB treatment outcome was analysed as a

dichotomous variable; we defined treatment outcomes as favourable if the patients were considered cured or completed treatment. Death, treatment failure, or loss to follow-up were all considered unfavourable outcomes [24]. When a patient has bacteriologically confirmed TB at the start of therapy, they are considered to have been cured of TB if a smear or culture-negative test result is established. Patients who finished the course of treatment without showing signs of failure but who did not have a sputum smear or culture findings documented are referred to as having completed treatment. Patients who experienced TB treatment failure had a positive sputum smear or culture test at five months or later during the course of the treatment. Patients who did not begin therapy or who had stopped treatment for at least two months consecutively are deemed lost to follow-up. A TB patient is deemed dead if they passed away for any cause prior to starting treatment or while in treatment [9].

**Exposure variables.**   We included the socio-demographic characteristics, comprising of age, sex, workplace, district, and region. We categorized and recoded regions in geographical zones as coastal, central, northern, western, southern highland, lake zone, and Zanzibar. The health facilities were recategorized as dispensaries, health centres, and hospitals. HIV status was determined by a positive or negative HIV test; anatomical TB sites were classified as pulmonary TB, extrapulmonary TB, or both. History of previous TB treatment was classified as per the NTLP TB treatment manual; these are new patient, relapse, treatment after loss to follow up, treatment after failure, and others.

TB regimens were categorized as recommended by WHO: Category I will consist of patients under RHZE for two months and subsequently Rifampicin and Isoniazid (RH) for four months. Patients with extrapulmonary TB, such as TB of the spine, joints, and bone, miliary TB, and TB meningitis, are prescribed RHZE for a period of 2 months and 10 months of RH, respectively. Conversely, category II of treatment for previously treated smear-positive patients (with no evidence of drug-resistant TB) will entail 3 months of RHZE and RHE for 5 months [25].

Similar to the national protocol, referral types were classified as self, community, and care and treatment centre (CTC) referrals and others. The diagnostic methods and findings were recorded as either sputum smear microscopy, culture, Gene Xpert, and/or chest X-ray. We categorised TB diagnostic methods as clinical TB diagnosis (Chest X-ray) and bacteriological TB diagnosis (smear microscopy, culture, or Gene Xpert). The mode of delivery of DOT was recorded as either facility-based or community-based DOT [25].

**Data management and analysis.**   Data management and analysis were conducted using Stata software version 17. All data, including dates, were encoded and converted into Stata-friendly formats. Missingness of the data accounted for less than 1% of most variables, and up to 82% of patients had complete treatment outcome data (Fig 1). We computed descriptive analysis using frequency and proportions for each independent categorical variable. The frequencies, proportions (%), and 95% confidence intervals (CI) were recorded for TB recurrence and unfavourable treatment outcomes. Pearson's chi-square test was used to compute bivariate analysis. We computed a multilevel multivariable mixed effects logistic regression model for TB recurrence and TB treatment outcomes to account for clustering at the facility level. We used a purposeful selection method to build the two models for TB recurrence and treatment outcomes. Age, sex, and HIV status were included in the multivariable model as apriori decided confounders based on prior knowledge [13,14,26,27]. Using the purposeful selection method, the multivariable model included apriori selected variables and variables with a p-value <0.2 [28] on bivariate analysis (*S1 and S2 Tables*). We reported an adjusted odds ratio and the corresponding 95% CI. A two-tailed p-value of <0.05 was considered statistically significant in the final models.

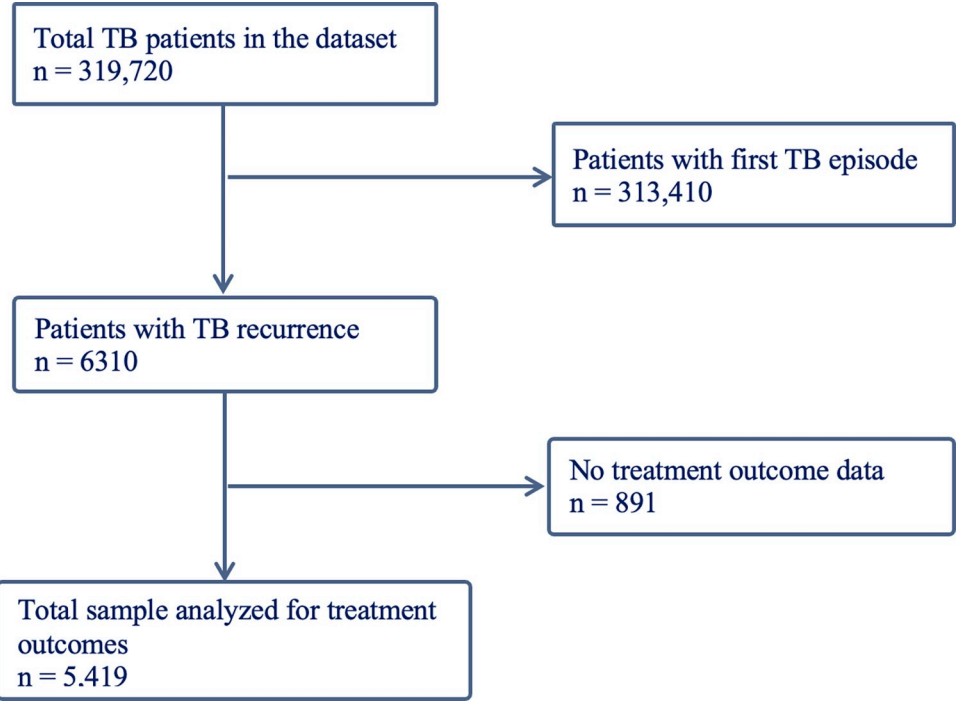

**Fig 1. Flowchart of patients included in the analysis.**

## Results

### Characteristics of study participants

A total of 319,720 participants were included in the study (Fig 1), with the majority being adults aged 25–49 (44.2%, n = 141,193) and above 50 years (31.6%, n = 101,039). About two-thirds were male (60.4%, n = 192,986), and half of the participants were from the coastal (29.3%, n = 93,577) and lake geographical zones (24.5%, n = 78,402). One-fifth of participants (22.8%, n = 72,396) were HIV positive, and more than three-quarters (79.4%, n = 253,690) of participants had pulmonary TB.

Half of the participants (53.5%, n = 170,996) were self-referred to the health facilities and were managed at the hospital facility level (46.2%, n = 147,653). The majority of participants were treated using the community-based DOT modality (98.2%, n = 302,476) and were under first-line TB treatment for pulmonary TB (98.6%, n = 310,813). About three-quarters of TB patients were diagnosed bacteriologically (72.7%, n = 207,734) (Table 1).

### Period prevalence and factors associated with TB recurrence

Newly diagnosed patients with TB made up the majority of participants (96.96%, n = 333,201), and about two in every hundred TB patients had a recurrent (relapse) TB episode (1.96%, n = 6,723) (Table 1).

After adjusting for both random and fixed effects at the facility level in the multivariable logistic model, age, sex, HIV status, TB referral types, facility level, geographical zones, DOT option, TB diagnostic methods, and treatment regimens remained as significant determinants of TB recurrence. The likelihood of TB recurrence was more than two times higher for patients aged 25 to 49 years (aOR = 2.33; 95% CI: 1.98–2.73; *p*-value < 0.001) and above 50 years (aOR = 2.81; 95% CI: 2.39–3.30; *p*-value < 0.001). Males had 40% extra odds of TB recurrence

**Table 1. Socio-demographic and facility-level characteristics of patients diagnosed with TB in Tanzania from 2018 to 2021 (N = 319,720).**

| Variables | n | % |
|---|---|---|
| **Age groups (n = 319,717)** | | |
| 0–14 | 48,878 | 15.3 |
| 15–24 | 28,607 | 8.9 |
| 25–49 | 141,193 | 44.2 |
| 50+ | 101,039 | 31.6 |
| **Sex (n = 319,718)** | | |
| Female | 126,732 | 39.6 |
| Male | 192,986 | 60.4 |
| **Disease classification (n = 319,604)** | | |
| Both | 780 | 0.2 |
| Extra pulmonary | 65,134 | 20.4 |
| Pulmonary | 253,690 | 79.4 |
| **HIV status[1] (n = 317,729)** | | |
| Negative | 245,132 | 77.1 |
| Positive | 72,396 | 22.8 |
| Unknown | 201 | 0.1 |
| **TB related referrals (n = 319,717)** | | |
| CTC | 33,010 | 10.3 |
| Community | 88,586 | 27.7 |
| Self-referrals | 170,996 | 53.5 |
| Others[2] | 27,125 | 8.5 |
| **Facility level (n = 319,720)** | | |
| Dispensary | 80,743 | 25.2 |
| Health Centre | 91,324 | 28.6 |
| Hospitals | 147,653 | 46.2 |
| **Geographical zones[3] (n = 319,720)** | | |
| Central | 40,091 | 12.6 |
| Coastal | 93,577 | 29.3 |
| Lake | 78,402 | 24.5 |
| Northern | 55,045 | 17.2 |
| Southern Highlands | 41,594 | 13.0 |
| Western | 8,025 | 2.5 |
| Zanzibar | 2,986 | 0.9 |
| **DOT option (n = 307,968)** | | |
| Facility | 5,492 | 1.8 |
| Community | 302,476 | 98.2 |
| **TB diagnostic method[4] (n = 285,877)** | | |
| Bacteriologically confirmed | 207,734 | 72.7 |
| Clinically diagnosed | 78,143 | 27.3 |
| **TB treatment regimen (n = 315,324)** | | |
| 2HRZE/10RH[5] | 4,049 | 1.3 |
| 2RHZE/4RH[6] | 310,813 | 98.6 |
| 2SRHZE/1RHZE/5RHE or 3RHZE/5RHE[7] | 462 | 0.1 |
| **History of treatment (n = 319,716)** | | |
| New | 309,895 | 96.9 |
| Other[8] | 1,793 | 0.6 |
| Recurrence | 6,310 | 2.0 |
| Treatment after failure patient | 440 | 0.1 |

(*Continued*)

**Table 1.** (Continued)

| Variables | n | % |
|---|---|---|
| Treatment after lost to follow up patient | 1,278 | 0.4 |
| **Year of TB diagnosis (n = 319,720)** | | |
| 2018 | 75,414 | 23.6 |
| 2019 | 82,056 | 25.7 |
| 2020 | 82,522 | 25.8 |
| 2021 | 79,728 | 24.9 |
| **DS-TB Treatment Outcome (n = 262,180)** | | |
| Completed treatment | 162,429 | 62.0 |
| Cured | 85,570 | 32.6 |
| Died | 10,944 | 4.2 |
| Lost to follow-up | 2,606 | 1.0 |
| Treatment failed | 631 | 0.2 |

CTC: Centre for Treatment and Care; DS-TB: Drug-sensitive TB; DOT: Directly Observed Therapy; TB: Tuberculosis; HRZE: Isoniazid, Rifampicin, Pyrazinamide, Ethambutol; SRHZE: Streptomycin, Rifampicin, Isoniazid, Pyrazinamide, Ethambutol.

[1]Patients with unknown HIV status excluded from analysis,

[2]Patients referred from inpatient department (IPD), outpatient department (OPD), diabetic clinic, voluntary counselling and testing (VCT), reproductive and child health clinics and others not defined

[3]Countries included in the geographical zones: Northern zone: Kilimanjaro, Tanga, Arusha, and Manyara. Coastal zone; Morogoro, Dar es Salaam, Pwani, Lindi, and Mtwara. Western zone: Katavi and Kigoma. Central zone: Tabora, Dodoma, and Singida. Lake zone: Kagera, Mwanza, Geita, Mara, Simiyu, and Shinyanga. Southern highlands zone: Songwe, Ruvuma, Mbeya, Njombe, Rukwa and Iringa. Zanzibar: Pemba and Unguja.

[4]TB diagnosis method: Bacteriologically confirmed diagnostic method: Gene Xpert, Microscopy and culture. Clinical diagnostic method: TB chart card and Chest X-ray.

[5]First line treatment regimen for extra-pulmonary TB.

[6]First line treatment regimen for pulmonary TB.

[7]Previous TB treatment regimen for patients with relapse TB or "Other" TB category before the new TB retreatment regimen introduced in 2019.

[8]Patients who were previously treated but with an unknown or undocumented outcome for the most recent treatment cycle

compared to females (aOR = 1.40; 95% CI: 1.31–1.49; *p*-value < 0.001). HIV-positive participants had 26% extra odds of TB recurrence compared to their HIV-negative counterparts (aOR = 1.26; 95% CI: 1.16–1.36; *p*-value < 0.001) and referral from CTC was associated with 38% additional risk of TB recurrence (aOR = 1.38; 95% CI: 1.20–1.58; *p*-value < 0.001). Bacteriological TB diagnosis (aOR = 1.61; 95% CI: 1.48–1.74; *p*-value < 0.001) was associated with a 61% additional risk of TB recurrence as compared to clinical TB diagnosis. Compared to community-based DOT, patients who received DOT at the facility had 12 times the odds of TB recurrence (aOR = 12.35; 95% CI: 11.13–13.71; *p*-value < 0.001). Patients receiving treatment from Zanzibar had about three-fold odds of TB recurrence compared to Southern Highlands (aOR = 2.71; 95% CI: 1.79–4.10; *p*-value < 0.001) and compared to 2018, patients diagnosed in 2019 (aOR = 1.16; 95% CI: 1.07–1.25; *p*-value < 0.001) and 2020 (aOR = 1.18; 95% CI: 1.09–1.28; *p*-value < 0.001) had 16% and 18% extra likelihood of having recurrent TB respectively. Further, compared to dispensary facilities, receiving treatment in hospital facilities was associated with 34% additional odds of TB recurrence (aOR = 1.34; 95% CI: 1.11–1.61; *p*-value = 0.002) (Table 2).

**Table 2. Factors associated with TB recurrence in Tanzania, from 2018 to 2021 (N = 273,807).**

| Variables | TB recurrence, N (%) | aOR | 95% CI | p-value |
|---|---|---|---|---|
| **Age group** | | | | **<0.001** |
| 0–14 | 261 (0.5) | 1 | | |
| 15–24 | 357 (1.2) | 1.29 | 1.06–1.56 | 0.010 |
| 25–49 | 3,354 (2.4) | 2.33 | 1.98–2.73 | <0.001 |
| 50+ | 2,338 (2.3) | 2.81 | 2.39–3.30 | <0.001 |
| **Sex** | | | | |
| Female | 1,896 (1.5) | 1 | | |
| Male | 4,414 (2.3) | 1.40 | 1.31–1.49 | <0.001 |
| **HIV status[1]** | | | | |
| Negative | 4,266 (1.7) | 1 | | |
| Positive | 1,994 (2.8) | 1.26 | 1.16–1.36 | <0.001 |
| **TB type** | | | | **<0.001** |
| Both | 12 (1.7) | 1 | | |
| Extra pulmonary | 714 (1.1) | 1.31 | 0.68–2.53 | 0.412 |
| Pulmonary | 5,583 (2.2) | 2.00 | 1.04–3.83 | 0.037 |
| **TB referrals** | | | | **<0.001** |
| Others[2] | 488 (1.8) | 1 | | |
| CTC | 1,006 (3.0) | 1.38 | 1.20–1.58 | <0.001 |
| Community | 1,523 (1.7) | 1.16 | 1.02–1.31 | 0.022 |
| Self-referral | 3,293 (1.9) | 1.16 | 1.04–1.30 | 0.011 |
| **Facility level** | | | | **0.004** |
| Dispensary | 1,428 (1.8) | 1 | | |
| HC | 1,838 (2.0) | 1.21 | 1.03–1.42 | 0.021 |
| Hospital | 3,044 (2.0) | 1.34 | 1.11–1.61 | 0.002 |
| **Geographical zones[3]** | | | | **<0.001** |
| Southern Highland | 565 (1.4) | 1 | | |
| Central | 770 (1.9) | 1.34 | 1.02–1.76 | 0.034 |
| Coastal | 2,485 (2.7) | 1.39 | 1.10–1.76 | 0.006 |
| Lake | 1,332 (1.7) | 1.24 | 0.98–1.57 | 0.078 |
| Northern | 887 (1.6) | 1.13 | 0.87–1.46 | 0.363 |
| Western | 191 (2.4) | 1.51 | 1.04–2.20 | 0.031 |
| Zanzibar | 80 (2.7) | 2.71 | 1.79–4.10 | <0.001 |
| **DOT Option** | | | | |
| Community | 4,925 (1.6) | 1 | | |
| Facility | 1,047 (19.1) | 12.35 | 11.13–13.71 | <0.001 |
| **TB diagnosis[4]** | | | | |
| Clinically diagnosed | 1,061 (1.4) | 1 | | |
| Bacteriologically confirmed | 5,046 (2.4) | 1.61 | 1.48–1.74 | <0.001 |
| **TB regimen** | | | | **<0.001** |
| 2RHZE/4RH[5] | 5,652 (1.8) | 1 | | |
| 2RHZE/10RH[6] | 117 (2.9) | 1.59 | 1.24–2.02 | <0.001 |
| 2SRHZE/1RHZE/5RHE[7] | 230 (49.8) | 9.66 | 7.49–12.51 | <0.001 |
| **Year of TB diagnosis** | | | | **<0.001** |
| 2018 | 1,814 (2.4) | 1 | | |
| 2019 | 1,657 (2.0) | 1.16 | 1.07–1.25 | <0.001 |
| 2020 | 1,630 (2.0) | 1.18 | 1.09–1.28 | <0.001 |

(Continued)

**Table 2.** (Continued)

| Variables | TB recurrence, N (%) | aOR | 95% CI | *p-value* |
|---|---|---|---|---|
| 2021 | 1,209 (1.5) | 1.04 | 0.95–1.14 | 0.364 |

aOR: adjusted Odds Ratio; CTC: Centre for Treatment and Care; DS-TB: Drug-sensitive TB; DOT: Directly Observed Therapy; TB: Tuberculosis; HRZE: Isoniazid, Rifampicin, Pyrazinamide, Ethambutol; SRHZE: Streptomycin, Rifampicin, Isoniazid, Pyrazinamide, Ethambutol.

[1] Patients with unknown HIV status excluded from analysis.

[2] Patients referred from inpatient department (IPD), outpatient department (OPD), diabetic clinic, voluntary counselling and testing (VCT), reproductive and child health clinics and others not defined.

[3] Countries included in the geographical zones: Northern zone: Kilimanjaro, Tanga, Arusha, and Manyara. Coastal zone; Morogoro, Dar es Salaam, Pwani, Lindi, and Mtwara. Western zone: Katavi and Kigoma. Central zone: Tabora, Dodoma, and Singida. Lake zone: Kagera, Mwanza, Geita, Mara, Simiyu, and Shinyanga. Southern highlands zone: Songwe, Ruvuma, Mbeya, Njombe, Rukwa and Iringa. Zanzibar: Pemba and Unguja.

[4] TB diagnosis method: Bacteriologically confirmed diagnostic method: Gene Xpert, Microscopy and culture. Clinical diagnostic method: TB chart card and Chest X-ray.

[5] First line treatment regimen for pulmonary TB.

[6] First line treatment regimen for extra-pulmonary TB.

[7] Previous TB treatment regimen for patients with relapse TB or "Other" TB category before the new TB retreatment regimen introduced in 2019

## Period prevalence and factors associated with unfavourable treatment outcomes among patients with recurrent TB

In every hundred patients with recurrent TB, ten patients had unfavourable treatment outcomes (9.58%, n = 519). Death was the most common unfavourable outcome recorded (Table 3).

After computing a multivariable logistic regression model, HIV status, geographical zones, DOT option, and TB diagnostic methods remained significant determinants of unfavourable treatment outcomes among patients with TB recurrence. The odds of having unfavourable treatment outcomes were significantly higher for HIV-positive participants, participants treated in the central and coastal zones and those receiving treatment in Zanzibar, those who were bacteriologically diagnosed, and those who received DOT at the facilities (Table 4). The odds of poor treatment outcomes were two-fold higher for participants receiving treatment at the central (aOR = 2.24; 95% CI: 1.33–3.78; *p*-value = 0.002) and coastal zones (aOR = 2.20; 95% CI: 1.40–3.47; *p*-value = 0.001) compared to northern zone. HIV-positive participants had 62% extra odds of unfavourable treatment outcomes compared to their HIV-negative counterparts (aOR = 1.62; 95% CI: 1.25–2.11; *p*-value < 0.001). Bacteriological TB diagnosis

**Table 3. Treatment outcomes for patients with drug susceptible recurrent TB (N = 5,419).**

| TB treatment Outcome | N (%) (95% CI) | TB treatment outcomes | N (%) |
|---|---|---|---|
| Favourable Outcomes | 4900 (90.4) 89.6–91.2 | Cured[1] | 2,387 (44.1) |
| | | Completed treatment[2] | 2,513 (46.4) |
| Unfavourable Outcomes | 519 (9.6) 8.8–10.4 | Died[3] | 322 (5.9) |
| | | Treatment failure[4] | 111 (2.0) |
| | | Loss to follow up[5] | 86 (1.6) |

[1] TB Patients who finished treatment with evidence of negative sputum bacteriology

[2] TB Patient completed treatment without evidence of negative bacteriology or treatment failure

[3] TB patients who died before starting treatment or during the course of treatment

[4] TB patients with positive sputum bacteriology after at least five months of treatment

[5] TB patients who had started treatment or the treatment was interrupted for two consecutive months or more

**Table 4. Factors associated with unfavourable treatment outcomes among patients with recurrent TB in Tanzania from 2018 to 2021 (N = 4,884).**

| Variables | Unfavourable Outcomes N (%) | aOR | 95% CI | p-value |
|---|---|---|---|---|
| **Age group (years)** | | | | **0.209** |
| 0–14 | 14 (5.9) | 1 | | |
| 15–24 | 35 (10.7) | 2.38 | 1.00–5.68 | 0.051 |
| 25–49 | 290 (10.1) | 2.22 | 1.01–4.87 | 0.048 |
| 50+ | 180 (9.1) | 2.34 | 1.06–5.18 | 0.036 |
| **Sex** | | | | |
| Female | 158 (9.8) | 1 | | |
| Male | 361 (9.5) | 1.05 | 0.83–1.32 | 0.693 |
| **HIV status[1]** | | | | |
| Negative | 288 (7.9) | 1 | | |
| Positive | 228 (12.9) | 1.62 | 1.25–2.11 | <0.001 |
| **TB types** | | | | **0.895** |
| Both | 0 (0.0) | 1 | | |
| Extrapulmonary | 47 (7.9) | 1.03 | 0.70–1.50 | 0.895 |
| Pulmonary* | 472 (9.8) | - | - | |
| **TB referral** | | | | **0.064** |
| CTC | 115 (12.9) | 1 | | |
| Community | 121 (10.0) | 1.17 | 0.81–1.68 | 0.398 |
| Self-referral | 238 (8.2) | 0.86 | 0.62–1.19 | 0.355 |
| Others[2] | 45 (11.2) | 1.27 | 0.80–2.00 | 0.309 |
| **Geographical zones[3]** | | | | **0.001** |
| Northern | 43 (6.0) | 1 | | |
| Central | 72 (10.4) | 2.24 | 1.33–3.78 | 0.002 |
| Coastal | 261 (11.5) | 2.20 | 1.40–3.47 | 0.001 |
| Lake | 82 (7.9) | 1.30 | 0.79–2.13 | 0.299 |
| Southern Highlands | 40 (8.6) | 1.41 | 0.80–2.48 | 0.239 |
| Western | 12 (7.4) | 1.06 | 0.43–2.65 | 0.895 |
| Zanzibar | 9 (11.7) | 3.49 | 1.39–8.75 | 0.008 |
| **DOT Option** | | | | |
| Community | 347 (8.2) | 1 | | |
| Facility | 122 (12.6) | 1.39 | 1.04–1.85 | 0.025 |
| **TB diagnosis[4]** | | | | |
| Clinically diagnosed | 66 (6.9) | 1 | | |
| Bacteriologically diagnosed | 437 (10.2) | 1.39 | 1.02–1.90 | 0.037 |
| **TB treatment regimen** | | | | **0.110** |
| 2RHZE/10RHE[5] | 7 (11.5) | 1 | | |
| 2RHZE/4RH[6] | 431 (8.8) | 0.48 | 0.20–1.14 | 0.095 |
| 2SRHZE/1RHZE/5RHE or 3RHZE/5RHE[7] | 33 (14.8) | 0.65 | 0.25–1.72 | 0.388 |
| **Facility level** | | | | **0.159** |
| Dispensary | 103 (8.5) | 1 | | |
| Health Centre | 107 (10.7) | 1.29 | 0.91–1.83 | 0.150 |
| Hospital | 245 (9.4) | 1.37 | 0.98–1.90 | 0.061 |
| **Year of TB diagnosis** | | | | **0.384** |
| 2018 | 192 (10.8) | 1 | | |
| 2019 | 141 (8.8) | 0.95 | 0.72–1.25 | 0.717 |
| 2020 | 137 (8.7) | 0.91 | 0.68–1.20 | 0.501 |

*(Continued)*

**Table 4.** (Continued)

| Variables | Unfavourable Outcomes N (%) | aOR | 95% CI | p-value |
|---|---|---|---|---|
| 2021 | 49 (10.5) | 1.26 | 0.85–1.87 | 0.242 |

aOR: adjusted Odds Ratio; CTC: Centre for Treatment and Care; DS-TB: Drug-sensitive TB; DOT: Directly Observed Therapy; TB: Tuberculosis; HRZE: Isoniazid, Rifampicin, Pyrazinamide, Ethambutol; SRHZE: Streptomycin, Rifampicin, Isoniazid, Pyrazinamide, Ethambutol.

*Observations dropped due to collinearity.

[1]Patients with unknown HIV status excluded from analysis.

[2]Patients referred from inpatient department (IPD), outpatient department (OPD), diabetic clinic, voluntary counselling and testing (VCT), reproductive and child health clinics and others not defined.

[3]Countries included in the geographical zones: Northern zone: Kilimanjaro, Tanga, Arusha, and Manyara. Coastal zone; Morogoro, Dar es Salaam, Pwani, Lindi, and Mtwara. Western zone: Katavi and Kigoma. Central zone: Tabora, Dodoma, and Singida. Lake zone: Kagera, Mwanza, Geita, Mara, Simiyu, and Shinyanga. Southern highlands zone: Songwe, Ruvuma, Mbeya, Njombe, Rukwa and Iringa. Zanzibar: Pemba and Unguja.

[4]TB diagnosis method: Bacteriologically confirmed diagnostic method: Gene Xpert, Microscopy and culture. Clinical diagnostic method: TB chart card and Chest X-ray.

[5]First line treatment regimen for extra-pulmonary TB.

[6]First line treatment regimen for pulmonary TB. [7]Previous TB treatment regimen for patients with relapse TB or "Other" TB category before the new TB retreatment regimen introduced in 2019

(aOR = 1.39; 95% CI: 1.02–1.90; p-value = 0.037) was associated with a 39% additional risk of unfavourable treatment outcomes as compared to clinical TB diagnosis. Compared to community-based DOT, patients who received DOT at the facility had 1.39 times the odds of poor treatment outcomes (aOR = 1.39; 95% CI: 1.04–1.85; p-value = 0.025) (Table 4).

## Discussion

This study sought to determine the extent and correlates of TB recurrence and poor treatment outcomes among patients with recurrent TB. About two in every hundred TB patients had a recurrent (relapse) TB episode. TB recurrence increased with age; with higher prevalence among participants aged 25 years and above. Males, older patients, HIV-positive participants, and those treated at the hospital facility level and under facility-based DOT had higher TB recurrence rates. Unfavourable treatment outcomes were recorded in almost 10% of patients with recurrent TB. The burden was higher among patients from Zanzibar, coastal and central geographical zones, people living with HIV, patients on facility-based DOT, and those with bacteriologically confirmed TB. Patients receiving treatment at the hospital level had marginally higher odds of unfavourable outcomes compared to those treated at the dispensary level.

The prevalence of recurrent TB among patients with susceptible TB in the current study resembles findings in other settings [7,29]. Meta-analytic pooled estimates reported an incidence of 5.6% TB relapse rate more than one year after treatment [29]. However, the burden is significantly higher in South Africa, which harbours the highest burden of HIV in the SSA region [30]. In 2019 and 2020, TB recurrence rates were significantly higher in Tanzania compared to 2018. Similarly, WHO reported increasing mortality and poor outcomes for TB patients globally during the same period; the impact was attributed to the COVID-19 pandemic that led to reduced access to TB diagnosis and treatment, leading to halted progress in TB elimination goals [31].

A number of other socio-demographic and clinical factors have been reported to predict the occurrence of TB recurrence [6,17]. Similar to the previous studies, being a male and of older age in the current study were associated with a higher risk of TB recurrence [6,32]. There has been a well-established link between older age and TB recurrence. Weaker immunity

among older adults explains the higher risk of TB reactivation and progression [32]. Moreover, with an increasing risk of other underlying diseases in older age, treatment adherence may be compromised, subsequently increasing the risk of disease recurrence [32]. A higher risk of recurrence in males may be related to relatively poor adherence and poor health-seeking behaviours, as reported previously [32].

The prevalence of poor outcomes (9.8%) for patients with TB recurrence was two times higher than among patients with new TB episodes (5.4%) in this study. Our study reported relatively lower rates of treatment outcomes than findings from other developing countries. In South Africa, lower cure rates were reported among patients in category II of TB treatment, with 20% in-hospital mortality and more than a quarter (26.4%) having poor outcomes [33,34]. In China, a five times higher rate of 56.1% poor treatment outcomes was reported among patients with drug-resistant recurrent TB [11]. Several factors may explain the higher rates of poor outcomes; TB drug resistance is the commonest among patients presenting with a recurrent TB episode and has been consistently linked with poor outcomes [11,34].

HIV–TB coinfection has been linked to a higher risk of TB recurrence and poor treatment outcomes for both patients with new or recurrent TB episodes [6,35]. Being habitant to 70% of the HIV burden globally, TB coinfection and subsequent poor outcomes in SSA are reported to be relatively higher [36]. We reported similar findings where HIV-positive patients with recurrent TB had almost twice the odds of TB recurrence and poor treatment outcomes compared to HIV-negative patients. Consistent with the current evidence, McGreevy et al. reported a lower success rate among HIV-infected patients on TB retreatment regimen in Haiti [37]. Relative higher rates of poor TB treatment outcomes were also reported among HIV-infected persons in Uganda [13]. Other evidence shows that HIV-infected recurrent TB patients were more likely to die than HIV-uninfected patients [37]. Living with HIV is associated with a number of general health complications; further increasing their risk of mortality, low cure rates, and unsuccessful TB treatment outcomes. HIV contributes to a higher likelihood of TB infection reactivation and progression resulting from impaired hosts' immune system [38]. Other contributors to poor outcomes include pill burden and subsequent poor adherence to treatment, higher risk of drug interactions, drug resistance, and adverse drug effects [39,40].

TB recurrence and poor treatment outcomes were significantly higher among bacteriologically diagnosed recurrent TB patients compared to those who were clinically diagnosed in this study. Patients' clinical presentation during an active TB infection can mimic a number of other clinical diagnoses; hence, in the absence of confirmatory tests, misdiagnoses are common [41,42]. Among patients in developing countries, this has contributed to delays in treatment for other potentially severe diseases, such as lung cancer [41–43]. It is possible that clinically diagnosed patients in our study may be receiving TB treatment while suffering from other diseases with similar presentations as TB, hence presenting with better TB outcomes. It was reported elsewhere that patients with TB recurrence are prone to have concurrent underlying comorbidities that contribute to poor outcomes regardless of the confirmatory TB diagnosis [33]. Considerations should be made for possible post-treatment monitoring for patients with recurrent TB without bacteriological results.

Almost all patients in our study were treated under community-based DOT: this is consistent with the evidence elsewhere [44,45]. Similar to our findings, community-based DOT conferred relatively higher cure rates than facility-based DOT in several other settings [45,46]. With community-based DOTs, treatment is delivered either at home or the workplace, according to patients' convenience. This is mostly favourable for both the patients and the overworked health systems in developing countries with high TB burden [45]. For this reason,

community / home-based DOT has been shown to promote treatment adherence and subsequently higher treatment success rates [44–46].

Our findings reported significantly higher proportions of poor treatment outcomes among recurrent TB patients in Zanzibar, Coastal, and Central geographical zones. This could be linked to a number of social-cultural and contextual factors that may vary across communities in Tanzania. Factors relating to the distribution of TB drug resistance may explain the higher rates of poor outcomes in the coastal and central zones [23]. Context-specific health system factors such as diagnostic methods and treatment delays, among others, may also explain the observed differences [47]. A study conducted in Zanzibar reported health system delays to be the main contributors to long delays in the TB treatment initiation [47]. This may explain the higher odds of poor treatment outcomes in Zanzibar despite having the lowest rates of HIV and TB drug resistance [23,48].

We reported relatively higher rates of poor outcomes among patients diagnosed and receiving treatment at the hospital facility level. Patients treated at hospitals may be referred with the most severe disease and with delays in treatment initiation due to faults and delays in the referral system [49]. However, contrary to our findings, some evidence shows higher treatment success rates among tertiary-level facilities [50,51]; this can be explained by the fact that these are well-equipped with relatively advanced diagnostic tools and specialized services.

The strength of this study lies in the fact that we analysed the national dataset with a large sample size, which enhances the generalizability of the results. The dataset had minimal missingness in most of the variables except a few; this could be related to the use of a digitalized system and strengthened monitoring and evaluation of the TB program. Our findings should be interpreted in the light of the following limitations. The first limitation is related to the use of secondary data, which limited the inclusion of other key factors associated with poor recurrent TB outcomes in the analysis. Variables relating to the time since the last TB episode, assessment of TB drug resistance [11,34], HIV-related factors on virological or immunological treatment failure, socio-economic factors, other comorbidities such as smoking, drug use [14], and chronic lung diseases [17,52] were not recorded in the dataset. For this matter, we acknowledge the possibility of residual confounding in our findings. Interpretation of our findings and programmatic implications are therefore focused on the individual and facility-level factors. The second limitation is inherent to the research design used for this study. With a cross-sectional design, causal inference could not be established between key determinants and TB recurrence and subsequent treatment outcomes. Further research that includes the key factors with comprehensive adjustment of the confounders and description of the causal pathway would be key to establishing the causal inference for factors associated with unfavourable outcomes among patients with recurrent TB.

There is, therefore, a need to address key issues in the management of recurrent TB to attain and maintain the national and overall WHO targets to end TB in Tanzania. While TB recurrence may be caused by TB reinfection with a different strain, TB relapse has been considered as a measure of TB treatment efficacy and, subsequently, the efficacy of the national TB programs [53]. We conclude that clinical monitoring and treatment of comorbidities, especially HIV, for patients with recurrent TB is crucial to improving treatment response. We propose further measures to improve outcomes, specifically for patients with HIV/TB coinfection. Drug susceptibility testing and close follow-up assessments may be considered, especially in the first year after treatment completion [38]. We reported over 18.3% of recurrent TB patients that were diagnosed clinically and had no reported bacteriological results. There is an urgent need to strengthen diagnostic methods to ensure the exclusion of TB resistance, especially among patients with relapse TB, which could contribute to suboptimal treatment and poor treatment outcomes. There is also a need to strengthen primary health facilities in TB

detection and management to reduce possible delays in treatment initiation. This can potentially influence early treatment and improve outcomes even for patients with severe diseases before their referral to hospital-level facilities. We also propose close follow-up for patients treated at the hospital facility level through appropriate DOT strategy to improve treatment outcomes.

## Supporting information

**S1 Table. Bivariate analysis of factors associated with TB recurrence in Tanzania from 2018 to 2021.**
(DOCX)

**S2 Table. Bivariate analysis of factors associated with unfavourable treatment outcomes among patients with recurrent TB in Tanzania from January 2018 to December 2021.**
(DOCX)

## Acknowledgments

We appreciate the support provided by the Monitoring & Evaluation unit at the National TB and Leprosy program, Ministry of Health, Tanzania. We acknowledge the support provided by Robert Balama in data curation. The authors would like to acknowledge the research support, training, and mentorship provided by the Division of Epidemiology and Biostatistics, School of Public Health, Faculty of Health Sciences, University of the Witwatersrand, Johannesburg, South Africa.

## Author Contributions

**Conceptualization:** Belinda J. Njiro, Riziki Kisonga, Joel Msafiri Francis.

**Data curation:** Belinda J. Njiro, Riziki Kisonga, Catherine Joachim, Galus Alfredy Sililo, Emmanuel Nkiligi, Joel Msafiri Francis.

**Formal analysis:** Belinda J. Njiro.

**Funding acquisition:** Belinda J. Njiro, Latifat Ibisomi, Tobias Chirwa.

**Methodology:** Belinda J. Njiro, Riziki Kisonga, Tobias Chirwa, Joel Msafiri Francis.

**Supervision:** Riziki Kisonga, Joel Msafiri Francis.

**Validation:** Riziki Kisonga, Joel Msafiri Francis.

**Visualization:** Belinda J. Njiro.

**Writing – original draft:** Belinda J. Njiro.

**Writing – review & editing:** Riziki Kisonga, Catherine Joachim, Galus Alfredy Sililo, Emmanuel Nkiligi, Latifat Ibisomi, Tobias Chirwa, Joel Msafiri Francis.

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
