## [Decision Letter · Decision Letter 0]

3 Jan 2024

Dear Dr Njiro,

Thank you very much for submitting your manuscript "Epidemiology and Treatment Outcomes of Recurrent Tuberculosis in Tanzania from 2018 to 2021 using the National TB dataset" for consideration at PLOS Neglected Tropical Diseases. As with all papers reviewed by the journal, your manuscript was reviewed by members of the editorial board and by several independent reviewers. The reviewers appreciated the attention to an important topic. Based on the reviews, we are likely to accept this manuscript for publication, providing that you modify the manuscript according to the review recommendations. 

Sincerely,

Victor S. Santos, Ph.D

Academic Editor

Mathieu Picardeau

Section Editor

Reviewer's Responses to Questions

**Key Review Criteria Required for Acceptance?**

**Methods**

-Are the objectives of the study clearly articulated with a clear testable hypothesis stated?

-Is the study design appropriate to address the stated objectives?

-Is the population clearly described and appropriate for the hypothesis being tested?

-Is the sample size sufficient to ensure adequate power to address the hypothesis being tested?

-Were correct statistical analysis used to support conclusions?

-Are there concerns about ethical or regulatory requirements being met?

Reviewer #1: The objective was clearly articulated in the hypothesis test, the study was designed well to address the objective and the population was clearly described with the test. The sample size sufficiently ensured the hypothesis and the analysis with good statistical support. the study was concerned with ethically approved

Reviewer #2: The authors set out to conduct a cross-sectional study of the determinants of treatment outcomes in recurrent TB. The hypothesis, results and statistical tests are represented clearly and tested appropriately. They have met the ethical and regulatory requirements.

Reviewer #3: Interesting study with clear and articulate objectives and hypothesis. But the study design is NOT a cross-sectional study but rather a retrospective cohort study with a secondary data analysis. The study population or participants for the study is not clear, is it general TB patients or TB recurrence patients? Looking at the title of the study. This lack of clarity is seen in the analysis outputs.

**Results**

-Does the analysis presented match the analysis plan?

-Are the results clearly and completely presented?

-Are the figures (Tables, Images) of sufficient quality for clarity?

Reviewer #1: Yes, the analysis was matched with the plan and the results were presented. Figure, table, and images are clearly stated

Reviewer #2: The analysis presented, matches the intent and the analysis plan laid out by the authors. The results are appropriately written. The authors have tested and presented the results mainly in the form of tables, which is an acceptable way for such data.

Reviewer #3: Wondering the choice of the study participants: The total TB patient in the data set (319,720) or the Patients with TB recurrence (6,310).

I thought the preliminary analysis (sociodemographic) should be focus on TB recurrence data set Not General TB dataset. The total dataset is not relevant, and will not bring out the silent features of the main participants in the study, such as gender difference among the TB recurrence patients etc.

The result looked at the determinants which is part of the epidemiology of TB recurrence, and the distribution such as prevalence 

I think N in the title of table 2 is 6,310 instead of N=273,807. Likewise, in Table 4 with N there should be 519 NOT 4,884.

**Conclusions**

-Are the conclusions supported by the data presented?

-Are the limitations of analysis clearly described?

-Do the authors discuss how these data can be helpful to advance our understanding of the topic under study?

-Is public health relevance addressed?

Reviewer #1: The conclusion is not separately stated in this research. it needs rewriting. The limitation of the analysis was clearly described. The authors described clearly the usefulness of the topic under the study. yes this work addressed the relevance of public health

Reviewer #2: The conclusions are supported by the data presented. However, some of the recommendations presented after the data analysis can be toned down. For example : "We recommend post-treatment follow-up and

prophylaxis with Isoniazid therapy or shorter regimen TB preventive therapy for patients with

recurrent TB, especially those coinfected with HIV." How will this be achieved is beyond the scope of analysis and interpretation. This relies on so many factors and it questions the feasibility and involves socio-poltical structures in place, which is not appropriate for the conducted study, such recommendations should solely be based on results presented.

Reviewer #3: The conclusions are supported by the data with the limitations stated clearly and succinct discussion of public health importance.

**Editorial and Data Presentation Modifications?**

Reviewer #1: Minor revision is there in this study grammar correction, and the conclusion part total not written they need modification

Reviewer #2: The authors have relied on data presentation with just tables, which is appropriate , I wonder if some other reviwers felt a bit more graphical depiction of some of the key data, would make it more appealing and easy for the readers.

Reviewer #3: Accept with minor revision

**Summary and General Comments**

Reviewer #1: This research was important in addressing information regarding awareness of TB in the study area also for globally. in section conclusion may need rewriting the other section is good.

Reviewer #2: The study is well conducted and well written. I have pointed out some areas of improvement but that should not prevent the editors from accepting the study for publication.

Reviewer #3: Interesting, large secondary data which highlight recurrent TB determinants, but would have depicted the sociodemographic features of TB recurrent patients instead of general TB patients in which the data was drawn.

PLOS authors have the option to publish the peer review history of their article (what does this mean?). If published, this will include your full peer review and any attached files.

Reviewer #1: Yes: Chimdesa Adugna

Reviewer #2: Yes: Abhimanyu Abhimanyu

Reviewer #3: No

Figure Files:

Data Requirements:

Reproducibility:

References

---

## [Editor Report · Decision Letter 1]

23 Jan 2024

Dear Dr Njiro,

Thank you very much for submitting your manuscript "Epidemiology and Treatment Outcomes of Recurrent Tuberculosis in Tanzania from 2018 to 2021 using the National TB dataset" for consideration at PLOS Neglected Tropical Diseases. As with all papers reviewed by the journal, your manuscript was reviewed by members of the editorial board and by several independent reviewers. The reviewers appreciated the attention to an important topic. Based on the reviews, we are likely to accept this manuscript for publication, providing that you modify the manuscript according to the review recommendations. 

Dear authors,

After carefully reviewing the re-submitted manuscript, I would like to request just one tiny modification before considering it acceptable for publication.

As you have designed a study to investigate the prevalence of TB recurrence and associated factors, your study is a "cross-sectional study". As you have designed a study to investigate the prevalence of TB recurrence and associated factors, your study is a "cross-sectional study". Please modify this throughout the text.

Sincerely

Prof. Victor S Santos

Academic Editor

PLoS Neglected Tropical Diseases.

Sincerely,

Victor S. Santos, Ph.D

Academic Editor

Mathieu Picardeau

Section Editor

Dear authors,

After carefully reviewing the re-submitted manuscript, I would like to request just one tiny modification before considering it acceptable for publication.

As you have designed a study to investigate the prevalence of TB recurrence and associated factors, your study is a "cross-sectional study". As you have designed a study to investigate the prevalence of TB recurrence and associated factors, your study is a "cross-sectional study". Please modify this throughout the text.

Sincerely

Prof. Victor S Santos

Academic Editor

PLoS Neglected Tropical Diseases.

Figure Files:

Data Requirements:

Reproducibility:

References

---

## [Editor Report · Decision Letter 2]

5 Feb 2024

Dear Dr Njiro,

We are pleased to inform you that your manuscript 'Epidemiology and Treatment Outcomes of Recurrent Tuberculosis in Tanzania from 2018 to 2021 using the National TB dataset' has been provisionally accepted for publication in PLOS Neglected Tropical Diseases.

Best regards,

Victor S. Santos, Ph.D

Academic Editor

Mathieu Picardeau

Section Editor

---

## [Editor Report · Acceptance letter]

12 Feb 2024

Dear Dr Njiro,

We are delighted to inform you that your manuscript, "Epidemiology and Treatment Outcomes of Recurrent Tuberculosis in Tanzania from 2018 to 2021 using the National TB dataset," has been formally accepted for publication in PLOS Neglected Tropical Diseases.

Best regards,

Shaden Kamhawi

co-Editor-in-Chief

Paul Brindley

co-Editor-in-Chief
